# Silencing of a Pectin Acetylesterase (PAE) Gene Highly Expressed in Tobacco Pistils Negatively Affects Pollen Tube Growth

**DOI:** 10.3390/plants12020329

**Published:** 2023-01-10

**Authors:** Greice Lubini, Pedro Boscariol Ferreira, Andréa Carla Quiapim, Michael Santos Brito, Viviane Cossalter, Maria Cristina S. Pranchevicius, Maria Helena S. Goldman

**Affiliations:** 1Departamento de Biologia, Faculdade de Filosofia, Ciências e Letras de Ribeirão Preto, Universidade de São Paulo, Ribeirão Preto 14040-901, SP, Brazil; 2PPG-Genética, Faculdade de Medicina de Ribeirão Preto, Universidade de São Paulo, Ribeirão Preto 14049-900, SP, Brazil; 3Departamento de Genética e Evolução, Universidade Federal de São Carlos, São Carlos 13565-905, SP, Brazil

**Keywords:** pollen–pistil interaction, plant fertility, plant reproduction, pistil development, fruit formation, pectin modification, *Nicotiana tabacum*

## Abstract

Successful plant reproduction and fruit formation depend on adequate pollen and pistil development, and pollen–pistil interactions. In *Nicotiana tabacum*, pollen tubes grow through the intercellular spaces of pistil-specialized tissues, stigmatic secretory zone, and stylar transmitting tissue (STT). These intercellular spaces are supposed to be formed by the modulation of cell wall pectin esterification. Previously we have identified a gene preferentially expressed in pistils encoding a putative pectin acetylesterase (PAE), named *NtPAE1*. Here, we characterized the *NtPAE1* gene and performed genome-wide and phylogenetic analyses of PAEs. We identified 30 PAE sequences in the *N. tabacum* genome, distributed in four clades. The expression of *NtPAE1* was assessed by RT-qPCR and *in situ* hybridization. We confirmed *NtPAE1* preferential expression in stigmas/styles and ovaries and demonstrated its high expression in the STT. Structural predictions and comparisons between NtPAE1 and functional enzymes validated its identity as a PAE. Transgenic plants were produced, overexpressing and silencing the *NtPAE1* gene. Overexpressed plants displayed smaller flowers while silencing plants exhibited collapsed pollen grains, which hardly germinate. *NtPAE1* silencing plants do not produce fruits, due to impaired pollen tube growth in their STTs. Thus, NtPAE1 is an essential enzyme regulating pectin modifications in flowers and, ultimately, in plant reproduction.

## 1. Introduction

Successful plant reproduction depends on appropriate anther and pistil development, compatible pollen–pistil interactions, and the signaling that allows directional pollen tube growth and double fertilization. Pollen–pistil interactions may differ depending on whether the species has a wet or dry stigma. Solanaceous species, such as *Nicotiana tabacum*, have wet stigmas, on which a sticky exudate is secreted at maturity [1,2]. When pollen grains land on a mature stigma, they first come into contact with the exudate that completely covers them. The pollen hydrates, and a pollen tube germinates and grows through the lipid-rich exudate that fills the intercellular spaces of the stigmatic secretory zone and stylar transmitting tissue [2,3,4]. These intercellular spaces are formed at the later stages of tobacco flower development toward anthesis [2] and are likely to occur by the modulation of cell wall pectin esterification [5].

Pectins are heterogeneous polysaccharides, present in the primary walls and the intercellular layer of plant cells. They may vary in molecular size, degrees of acetylation, and methylation depending on their plant source. Pectins are assumed to have an important function in cell-cell adhesion [6], which occurs at several steps during plant reproduction: when pollen grains land on a stigma when pollen tubes traverse the transmitting tract of the style, and when a sperm cell meets the egg cell during fertilization. Cell adhesion and intercellular communication are modulated by the degree and pattern of pectin esterification, which results mainly from the action of the enzymes pectin acetylesterase (PAEs–EC 3.1.1.6) and pectin methylesterases (PMEs–EC 3.1.1.11).

Several studies on PMEs, carried out in recent years, have increased the knowledge about the role of these enzymes. PMEs can remove methyl groups from pectin, promoting calcium-dependent gelation [7]. On the other hand, PMEs can have, sometimes, the opposite effect, resulting in cell wall loosening. Thus, it is necessary to properly regulate PME activity during plant processes, achieved by pectin methylesterase inhibitors (PMEIs) [7]. In the *Arabidopsis thaliana* genome, there are 66 pectin methylesterase (PME) and 71 pectin methylesterase inhibitor (PMEI) genes [8,9,10]. PMEs are involved in different processes such as plant growth [11,12], and plant defense responses against biotic [13,14] and abiotic stresses [15,16,17]. In plant reproduction, PMEs have roles, for example, in pollen grain germination [18] and the regulation of pollen tube growth [19,20,21]. They are also involved in preventing the entrance of supernumerary pollen tubes into ovules, which is mediated by FERONIA receptor kinase [22]. This beautiful work from Alice Cheung’s group has demonstrated that *A. thaliana pme34* and *pme44* mutants, as well as *PMEI1* overexpression, result in multiple pollen tubes per ovule [22]. In maize (*Zea mays*), PMEs and their PMEI counterpart genes are involved in anther and pollen development [10] and have been associated with cross-incompatibility systems, controlling the decision between pollen acceptance versus rejection [23]. In the tomato (*Solanum lycopersicum*) genome, there are 57 non-redundant PME genes [24]. The analysis of gene expression of the PMEs genes in the different parts of the tomato plant suggests they fulfill different roles, including vegetative organ development [25] and fruit softening through ripening [24]. In the tobacco (*N. tabacum*) genome, there are 121 PME genes and they play multiple roles in regulating plant development and responses to various stresses [26]. Overexpression of the *NtPME029* gene was shown to promote root development, while the overexpression of the *NtPME043* gene resulted in enhanced tolerance to salt stress in tobacco [26].

In this context, the diverse biological functions of PMEs have been comprehensively explored in recent years. In contrast, despite the importance of the level of pectin acetylation in cell wall properties and signaling, PAEs and their roles are still underexplored [7,27]. Similar to methyl esters, it is assumed that the degree of pectin acetyl esters contributes to the regulation of the polymer’s biophysical properties [28], influencing their gelation characteristics, solubility, and digestibility [29]. The level of pectin acetylation is determined by: (1) the addition of acetyl-esters in the Golgi complex during pectin synthesis; and (2) the removal of acetyl esters by acetylesterase [30]. To our knowledge, a pectin acetylesterase inhibitor, such as the PMEI for pectin methylesterification, has not been identified to date.

While genome-wide PAE family analyses are available for *A. thaliana* [27] and *Populus trichocarpa* [31], only a few plant PAEs are functionally characterized. In *A. thaliana*, *PAE8* and *PAE9* were studied, and their mutants displayed reduced inflorescence growth, demonstrating the importance of pectin acetylation in plant growth and development [30]). The *A. thaliana pae11* mutant exhibited reduced photosynthesis associated with the altered cell wall composition [32]. The *Vigna radiata VrPAE1* gene, one of the first cloned PAE sequences [33], was expressed in transgenic potato tubers, resulting in a stiffer potato tuber tissue and a stronger cell wall matrix due to the de-acetylation [34]. PAEs are also involved in biotic stress, as in the citrus bacterial canker (CBC) caused by *Xanthomonas citri* infection [35]. In *Phaseolus vulgaris*, the *PvPAE8* gene is specifically expressed in the seed coat of developing seeds [36]. The lack of a functional *PvPAE8* gene was associated with an increased rate of water absorption by the seed and a higher percentage of aged seed germination. The authors suggest that the decreased acetylation of pectin leads to enhanced interaction with Ca^2+^, contributing to water impermeability [36]. The *PtPAE1* gene of *P. trichocarpa* was cloned and demonstrated to have pectin de-acetylation activity in vitro [29]. Interestingly, overexpression of *PtPAE1* in transgenic *N. tabacum* plants impaired the cellular elongation of floral styles and filaments, the germination of pollen grains, and the growth of pollen tubes [29].

In a previous macroarray study, we identified a cDNA clone (TOBEST clone S004A06) encoding a putative PAE as preferentially expressed in the stigma/style of *N. tabacum* [37]. The corresponding gene was named *NtPAE1* and is characterized here. It belongs to a family of sequence conserved PAEs, in a clade of genes preferentially expressed in flowers. The NtPAE1 protein has a structure very similar to functional carboxylesterases, therefore, confirming its identity as a pectin acetylesterase. Our results show that the overexpression of *NtPAE1* affects flower growth, reducing the final size of floral organs. Meanwhile, its silencing disturbs pollen formation and negatively influences pollen tube growth. Plants with reduced levels of *NtPAE1* also do not provide an appropriate pistil path for pollen tube growth and, thus, do not produce fruits by self- or cross-pollination. Our study contributes to the understanding of the role of pectin-modifying enzymes, more specifically of PAEs, in the reproductive process of plants.

## 2. Materials and Methods

### 2.1. Plant Material

Wild type and transgenic *N. tabacum* L. cv Petit Havana SR-1 plants were grown on Bioplant^®^ substrate with vermiculite, irrigated from below, and cultivated in a growth chamber (Weiss–Gallenkamp, 55% humidity, 16-h day/8-h night regime, 22 °C). After initial growth plants were transferred to a greenhouse under standard conditions in Ribeirão Preto–SP, Brazil (21°10′24″ S 47°48′24″ W, with an average temperature of 22 °C in winter and 27 °C in summer), with daily irrigation by automatic sprinkler, for 15 min every 12 h.

### 2.2. RNA Extractions and RT-qPCR

Samples for RNA extractions were collected from young plants (roots, stems, and leaves) and the flowers of adult plants. The samples of floral organs (sepals, petals, stamens, stigmas/styles, and ovaries) represent pools of different developmental stages as defined by Koltunow et al. [38]. Furthermore, for a more detailed analysis of expression during flower development, stigmas/styles and ovaries were collected at each of the 12 stages of flower development (stage 1 has the floral organs fully differentiated, including pistils, and tetrads are present at anthers; at stage 12 anther dehiscence and anthesis occur; for more details of each stage see Koltunow et al. [38]). Samples were immediately frozen in liquid nitrogen and stored at −70 °C for later RNA extractions. All RNA extractions were made with Plant RNA Reagent (Invitrogen, Waltham, MA, USA), according to the manufacturer’s instructions. Total RNA was treated for DNA contamination with RQ1 DNAse (Promega, Madison, WI, USA), followed by cDNA synthesis with Superscript III Reverse Transcriptase (Invitrogen, Waltham, MA, USA). The genes *GLYCERALDEHYDE-3-PHOSPHATE DEHYDROGENASE* (*GAPDH*) and *ACTIN* were used as reference genes for comparisons in different organs, while only *ACTIN* was used for stigmas/styles and ovaries (primer information in Appendix A). The primer pair RT-PAE-Fw and RT-PAE-Rv was used to detect *NtPAE1* expression (Appendix A). The RT-qPCR reactions were carried out in three technical replicates with the SYBR Green Universal PCR Master Mix (Applied Biosystems, Warrington, UK) and analyzed on the ABI Prism 7700 Sequence Detection System (Applied Biosystems). Expression levels were calculated by the ΔΔCt method, with exception of the expression in stigmas/styles during development, which was calculated by the standard curve method.

### 2.3. In Situ Hybridization

Stigmas and styles at defined stages of flower development [38] were analyzed as described by Jackson [39]. Sense and antisense riboprobes were prepared by in vitro transcription of the TOBEST clone S004A06, with the DIG RNA Labeling Kit SP6/T7 (Roche, Mannheim, Germany). Digoxigenin (DIG)-labeled probes were detected using anti-DIG antibody conjugated with alkaline phosphatase (AP) (Roche, Mannheim, Germany) and the OneStep NBT/BCIP Detection Mixture (Thermo Fisher Scientific, Rockford, IL, USA). The slides were photographed with an Axiolab microscope (Zeiss, Jena, Germany) equipped with an AxioCam Color camera (Zeiss, Jena, Germany).

### 2.4. Cloning Procedures

The *NtPAE1* CDS was amplified from the TOBEST clone bank [37] using oligonucleotides PAE-attB1 and PAE-attB2 (Appendix A) and cloned into the pCR^®^2.1–TOPO^®^ vector by TOPO-TA cloning (Invitrogen). The product was re-amplified with a sequential PCR using BP1 and BP2 oligonucleotides (Appendix A) to generate complete attB sites for use in recombination with a pDONR201 vector (Invitrogen, USA). The primer pair PAE-attB1 and Ri-PAE-attB2 (Appendix A) was used to amplify an 800 bp sequence (nucleotides 98-897 of the mRNA—Appendix A) for the RNAi cassette (Ri-NtPAE1). Vectors pK7WG2 and pK7GWIWG2 were used to clone *NtPAE1* and *Ri-NtPAE1* sequences for overexpression and RNAi silencing, respectively, using the Gateway System (Invitrogen, Waltham, MA, USA).

### 2.5. Plant Transformation

Plant materials used in genetic transformations were prepared by sowing surface-sterilized seeds of *N. tabacum* cv Petit Havana SR-1 on solid Murashige–Skoog (MS) medium, which were placed in a growth chamber at controlled conditions (26 °C temperature and 16 h light/8 h dark photoperiod). Young seedlings were individualized in sterile glass jars containing fresh MS medium and kept in the same growth conditions until transformation. On the day of transformation, leaf disks were co-cultivated with *Agrobacterium tumefaciens* C58C1RifR (pGV2260) cultures containing the appropriate constructions and regenerated in vitro according to a previously described protocol [40].

### 2.6. Scanning Electron Microscopy (SEM)

Closed anthers at stage 11 were fixed in formalin–acetic acid–alcohol (FAA) solution [41], submitted to vacuum (15 mmHg) for 15 min and kept in fixative solution at RT for 24 h. Subsequently, the material was dehydrated in an ethanol series (50% to 100%) twice and for 2 h each, then dried to a critical point with a CPD 030 (BAL-TEC) in CO_2_. The dried material was covered with a 15 nm gold layer in a SCD 050 metallizer (BAL-TEC). The samples were analyzed under 15 kV voltage, and the electron micrographic record was performed using a scanning electron microscope JSM-5200 (JEOL) and a photographic camera JEOL MP.

### 2.7. Pollen Tube Growth Analyses

Controlled self- and cross-pollination were performed with mature pollen grains of wild type and PAERi-16.2 on emasculated stage 11 flowers [38]. Then, stigmas and styles were collected 7 h post-pollination, longitudinally hand sectioned, mounted on glass slides containing 0.05% aniline blue solution [42], and carefully squashed between the glass slide and coverslip before imaging. The samples were photographed under a KS-300 Imaging System (Zeiss).

### 2.8. Bioinformatics Analyses

#### 2.8.1. Sequence Retrieval

Amino acid sequences of PAEs from *A. thaliana* and *P. trichocarpa* were retrieved according to [27] and [31], respectively. Putative PAEs from *N. tabacum* and *S. lycopersicum* were obtained by BLASTP searches using *A. thaliana* and *P. trichocarpa* PAEs as queries from the reference tobacco and tomato annotations *Nitab4.5* and *ITAG4.0* (solgenomics.net), respectively. The top five best hits for each query were selected for further analysis following duplicate sequence removal. The amino acid sequences of *P. vulgaris* PAE8 (Phytozome accession Phvul.003g277600.1), and *V. radiata* PAE1 (NCBI accession X99348) were retrieved from Palmer et al. [36] and Breton et al. [33], respectively. The amino acid sequence of NtPAE1 was obtained by translation of its sequenced CDS, which was submitted to GenBank under accession number OQ091254.

#### 2.8.2. Phylogenetic Analyses

The nonredundant set of PAE sequences was aligned with the MAFFT v7.487 software [43], with parameters *“–allowshift–unalignlevel 0.1–bl 80–maxiterate 1000–globalpair"*. The resulting alignment was input to the SMS v2.0 software [44], followed by a Maximum Likelihood phylogenetic tree reconstruction with the PhyML v3.3.20220408 software [45] with parameters *“-b 1000 -f o -v 0.083 -m custom -a 1.040 -o tl”*.

#### 2.8.3. Mining of Expression Data

Expression data for candidate PAE genes corresponding to the amino acid sequences of the tree were obtained from high-throughput experiments of *A. thaliana* [46], *P. trichocarpa* (ePlant-Poplar), *P. vulgaris* [47], *S. lycopersicum* (ePlant-Tomato), and *N. tabacum* [48]. Since the expression values of each species were given in different units of measurement, we applied a per-gene z-score transformation strategy to the data, to allow for the comparison of the expression variation between the chosen tissues. The z-score data were plotted in a matrix and associated with the phylogenetic tree with the *ggtree* 3.4.1 package [49] in R 4.2 [50]. The complete dataset is available in Appendix A.

#### 2.8.4. Structure Prediction and Structural Alignment

Three-dimensional structures for NtPAE1, VrPAE1, and PvPAE8 were predicted with the ColabFold software [51] in the AlphaFold2-MMseqs2 mode with default parameters. The best ranking model, with the highest average pLDDT (predicted Local Distance Difference Test—[52,53]) was chosen for further analysis. Individual prediction metrics are available in Appendix A. The best-ranking models were aligned to a reference *Homo sapiens* NOTUM protein structure (PDB 6R8Q) with the MatchMaker tool [54] from the ChimeraX software [55]. Pairwise structural alignment scores were calculated with TM-align v20190822 [56] to assess sequence-independent structural similarity. Sequence representation of structural alignments was made by aligning the predicted structures with the PROMALS3D software [57].

## 3. Results

### 3.1. NtPAE1 Encodes a Flower-Specific Pectin Acetylesterase Preferentially Expressed in Pistils

The cDNA clone TOBEST S004A06 was completely sequenced and shown to be 1485 bp long (Appendix A). It has a 5${}^{\prime }$-UTR of 97 bp and a 3${}^{\prime }$-UTR of 180 bp before the polyA-tail (Appendix A). This cDNA contains the full-length coding sequence of a protein of 392 amino acids, which contains a putative signal peptide (probability of 0.9821) with a cleavage site between amino acids 28 (Glycine) and 29 (Aspartate), predicted by SignalP6 [58]. This sequence corresponds to the *N. tomentosiformis*-derived gene present in the *N. tabacum* allotetraploid genome and was named *NtPAE1* (scaffold Nitab4.5_0000063, position 1,512,252 to 1,519,956 of the *N. tabacum* genome at Sol Genomics Network—gene accession Nitab4.5_0000063g0300, Appendix A). This gene has 13 exons and produces two transcripts (Nitab4.5_0000063g0300.1, coding for 392 amino acids and Nitab4.5_0000063g0300.2 corresponding to 314 amino acids). The alternative transcript Nitab4.5_0000063g0300.2 is produced by an alternative 3${}^{\prime }$ splicing site (A3′SS) at the beginning of exon 12 (Appendix A), resulting in a smaller protein due to the introduction of a premature termination codon (PTC—142 nucleotides upstream of the original stop codon). The protein encoded by transcript Nitab4.5_0000063g0300.1 contains the complete PAE domain (PFAM accession pfam03283) spanning residues 29-376, which is almost the entire protein sequence (Appendix A), and a predicted phosphorylated serine at position 356 (0.634—MusiteDeep analysis for protein post-translational modification, [59]). The alternative transcript does not produce the complete PAE domain and is, presumably, subject to nonsense-mediated mRNA decay (NMD).

To confirm the tissue expression of *NtPAE1*, we performed a detailed RT-qPCR analysis in the different vegetative and reproductive organs of *N. tabacum*—roots, stems, leaves, sepals, petals, stamens, ovaries and stigmas/styles. *NtPAE1* is expressed exclusively in floral organs, except sepals, with higher expression levels in stigmas/styles and ovaries (Figure 1A). In situ hybridization of stage 11 stigma/style longitudinal sections show intense probe concentration in the stylar transmitting tissue, the path used by the pollen tubes growing towards the ovules in the ovary (Figure 1B, arrow).

During development, the *NtPAE1* gene has an intermediate expression level in the stigmas/styles at the first three developmental stages after pistil is fully differentiated [38], decreasing at stage 4 and being relatively low at stages 4 to 9 (Figure 1C). The expression level increases from stage 10 toward anthesis (stage 12), a period in which the stigma and style are in preparation to receive the pollen grains. The highest *NtPAE1* expression level in stigmas/styles is reached at stage 12, when anthesis and pollination occur. In ovaries (Figure 1D), *NtPAE1* is highly expressed at the earlier stages of flower development (stages 1 to 4), during which ovules are formed [60]. These results suggest that well-controlled pectin de-esterification by PAEs is important during pistil developmental processes.

### 3.2. Genome-Wide Identification of PAE Encoding Genes in N. tabacum

To identify the genes encoding PAEs in the *N. tabacum* genome, we used as queries the 12 PAE amino acid sequences identified in *A. thaliana* [27], and the 10 PAE sequences of *P. trichocarpa* [31], to our knowledge the only two species with genomes fully analyzed for PAE sequences so far. We found 30 non-redundant putative PAE sequences in the *N. tabacum* genome v4.5 [61] by similarity searches with BLASTP 2.13.0+ [62], including NtPAE1 here characterized. The remaining 29 PAE sequences in the *N. tabacum* genome were named *NtPAE2* through *NtPAE30* according to their positions in the chromosomes or scaffolds (Appendix A). All identified sequences have complete PAE domains, as predicted by CD-Search [63] (Appendix A). Signal peptides were predicted only in 14 of the 30 sequences (NtPAE1, NtPAE3, NtPAE5, NtPAE7, NtPAE9, NtPAE11, NtPAE13, NtPAE15, NtPAE17, NtPAE22, NtPAE23, NtPAE24, NtPAE25 and NtPAE26—Appendix A). The latter sequences, except for NtPAE11, also have the triad S–D–H (serine, aspartate, and histidine), predicted to be the active site of the PAE domain [27].

To expand the analyses of the PAE genes in Solanaceae, we also searched the sequences present in the diploid *S. lycopersicum* genome. We found 16 non-redundant putative PAE sequences, a number similar to the ones found in the *A. thaliana* and *P. trichocarpa* genomes. These results suggest that the higher number of PAE sequences in the *N. tabacum* genome may be due to its allotetraploid nature as a hybrid between *N. sylvestris* and *N. tomentosiformis* [48,61,64].

To further investigate *NtPAE1* and its relation to the other plant PAEs, we used the aforementioned PAE amino acid sequences to construct a maximum likelihood phylogenetic tree. On this tree, we also included two PAE sequences that have been functionally characterized and described in the literature, from *P. vulgaris* (PvPAE8, [36]) and *V. radiata* (VrPAE1, [33]). The resulting phylogenetic tree (Figure 2) has a similar topology to the trees produced by Gou et al. [29] and by Phillippe et al. [27], concerning clades I to III. Additionally, 5 out of 6 functionally characterized PAEs (see below) are grouped in clade III (Figure 2). Interestingly, a new clade (IV) was formed, containing exclusively Solanaceae sequences, including NtPAE1. In total, clade IV has 19 *N. tabacum* and 10 *S. lycopersicum* PAEs, which correspond to 63.3% and 62.5% of their PAE family proteins, respectively. The remaining Solanaceae sequences are distributed along the three distinct clades, among members of the other species.

As a next step in the characterization of the PAE gene family in *N. tabacum*, we evaluated the expression of the 30 putative PAEs and the PAEs from the other species by analysis of public RNA-seq data of *N. tabacum* [48], *A. thaliana* [46], *P. trichocarpa*, *P. vulgaris* [47], and *S. lycopersicum* (Appendix A). It is necessary to note that the expression matrix was built by comparing only the information available for most of the genes, corresponding to the four tissues shown in Figure 2 (roots, young leaves, mature leaves, and flowers) and, therefore, represents a limited assessment. Our results demonstrate a tissue-preferential expression for groups of PAEs and distinct patterns among clades (Figure 2). Moreover, some PAEs are generally more expressed in comparison to other members of the family (Appendix A). Remarkably, all members of PAE clade IV, in which NtPAE1 is included, have preferential expressions in flowers, highlighting the importance of PAEs in the Solanaceae flower development and the reproductive process.

### 3.3. NtPAE1 Is Structurally Very Similar to Functionally Characterized PAEs and Other Carboxylesterases

So far, few plant PAEs have been functionally characterized, and they are PAE8, PAE9, and PAE11 in *A. thaliana*, PtPAE1 in *P. trichocarpa* [29], VrPAE1 in *V. radiata* [33,34], and PvPAE8 in *P. vulgaris* [36]. To evaluate whether NtPAE1 shares structural properties with enzymatically active PAEs, we performed de novo protein folding predictions of NtPAE1, VrPAE1, and PvPAE8 (Figure 3 and Appendix A). Next, the best-ranking models of these PAEs were aligned to an X-ray diffraction structure of the *H. sapiens* carboxylesterase NOTUM, in complex with its inhibitor benzotriazole [65]. The superposed structures have nearly identical three-dimensional positionings of the predicted active site residues serine, aspartate, and histidine (the triad S-D-H), suggesting strong functional homology (Figure 3 and Appendix A). We additionally evaluated fold similarity by pairwise comparisons of the structural models with TM-align [56]. In every comparison, the alignment score is higher than 0.5 (Appendix A), demonstrating that the assessed sequences assume the same fold as the carboxylesterase HsNOTUM and confirm NtPAE1 identity as an enzymatically active PAE.

### 3.4. Altered Expression of the NtPAE1 Gene Affects Flower Development and Fertility

To evaluate the role of NtPAE1 in flower development and plant reproduction, we produced transgenic *N. tabacum* plants overexpressing *NtPAE1* and plants with RNAi-mediated silencing of *NtPAE1*. Morphological analyses show that at least two (Ov8.1 and Ov10.1) out of 15 independent overexpression transgenic lines have thinner pistils (Figure 4A). All overexpression transgenic lines have smaller flowers compared to the wild type (Figure 4(Aa,Ab)). In parallel, RNAi transgenic lines seem to have pistils that are harder and firmer than the wild type, especially line Ri16.2 (Figure 4A), which has the lowest *NtPAE1* transcript level. Additionally, the pollen grain morphology of RNAi plants revealed aberrant characteristics in comparison to wild type pollen, including malformed walls and apertures (Figure 4B).

To assess the consequences of lower *NtPAE1* transcripts in plant reproduction, we performed controlled self- and cross-pollination in the transgenic line Ri16.2 and compared it with wild type plants. Pollen tube growth was assayed by aniline blue staining, 7 h after controlled pollination. Wild type pollen tubes showed normal growth in wild type pistils (Figure 5(Aa,Ba)), reaching long distance post-pollination (Figure 5(Aa)). However, in Ri16.2 pistils, the wild type pollen tubes reached a shorter distance after the same time (Figure 5(Ad)). When pollen tubes from Ri16.2 transgenic plants were grown on Ri16.2 pistils, very few pollen grains germinated (Figure 5(Bb)) and the few growing pollen tubes displayed the shortest growth, reaching only the initial portion of the stylar transmitting tissue (Figure 5(Ab)). Meanwhile, a few pollen tubes from Ri16.2 transgenic plants were able to grow slightly further into wild type pistils (Figure 5(Ac)). Therefore, transgenic plants Ri16.2 were unable to produce fruits and seeds after natural or controlled pollination (Figure 5C), demonstrating their infertility. It should be mentioned that overexpression plants were fertile, despite producing smaller fruits with fewer seeds than wild type plants.

To further investigate the stylar transmitting tissue effect on pollen tube growth, we performed scanning electron microscope analyses of longitudinal sections of the styles of transgenic (Ov10.1 and Ri16.2) and wild type plants (Figure 6). It became evident that there is greater adhesion between cells in the stylar transmitting tissue of silenced plants for *NtPAE1* gene (Ri16.2), compared to the wild type which, in turn, has a better structured stylar transmitting tissue than the *NtPAE1* overexpression plants.

## 4. Discussion

Pectin and pectin esterification are related to cell wall mechanical properties and are involved in various cellular processes, including morphogenesis and development [7]. Despite the important contribution of PAEs in the control of pectin esterification, only a few PAE genes have been characterized so far. This work presents the characterization of *NtPAE1*, the first pectin acetylesterase to be studied in Solanaceae. We achieved a genome-wide identification of PAE sequences in *N. tabacum* and *S. lycopersicum* and performed a phylogenetic analysis including the PAE sequences from *A. thaliana* and *P. trichocarpa*, the two species with genomes completely surveyed for PAEs [27,31]. In the *N. tabacum* genome, we found 30 non-redundant putative PAE sequences, which is approximately two times the number of genes present in the other genomes analyzed. However, only 14 of these genes encode a signal peptide, and from these, only 13 PAEs have all the characteristics expected for functional enzymes. The remaining sequences might represent incompletely sequenced PAEs, pseudogenes, or even neofunctionalization after the interspecific hybridization that gave origin to the *N. tabacum* genome.

Our analysis revealed an additional clade (clade IV), not described in the previous works [27,31], containing exclusively Solanaceae PAEs. Interestingly, PAE clade IV members, including NtPAE1, are preferentially expressed in flowers (Figure 1 and Figure 2). Are clade IV PAE sequences related to specific functions in wet stigma species? The evolutionary history of dry and wet stigmas is not clear, and the distribution of the stigma types is extremely unpredictable [66]. Some families have representatives with both types of stigmas, indicating that the stigma type (wet, dry, or even semi-dry) may evolve through a continuous process of gain and loss during angiosperm diversification [66]. *A. thaliana* is a species of dry stigmas, while the *Populus* genus has several species with wet stigmas. The fact that the completely sequenced *P. trichocarpa* genome has no PAE members in clade IV suggests this is not a common feature or a prerequisite of wet stigma species. Therefore, it is possible that these PAE sequences are associated with flower development and plant reproduction processes occurring in species of the Solanaceae family. More studies with PAE genes in other species will be necessary to clarify this point.

In this work, we have shown that the predicted three-dimensional structure of NtPAE1 is over 92% similar to the enzymatically active PAEs PvPAE8 and VrPAE1 (Figure 3, Appendix A). Additionally, all three NtPAE1, PvPAE8, and VrPAE1 are at least 86% similar in structure to the *H. sapiens* NOTUM carboxylesterase, a PAE homolog [27,31]. These structural comparisons of NtPAE1 with functionally well-characterized pectin acetylesterase confirmed the enzymatic activity of the NtPAE1 protein.

Immunocytochemical investigations in unpollinated pistils of *Petunia hybrida* (a wet stigma species) demonstrated high levels of de-esterified pectin [67,68], which reinforces the importance of the activity of PAEs and PMEs in modifying pectin from specialized tissues of the stigma/style. Additionally, it has already been reported that PMEs of the transmitting tissue act directly on the modification of the cell wall of pollen grains, limiting the germination of pollen grains of other species [69]. Interestingly, the highest NtPAE1 expression level in stigmas/styles is reached at stage 12, when anthesis and pollination occur. This higher NtPAE1 expression level in stigmas/styles may be due to its induction by pollination and/or a role similar to what has been described for PMEs. Another possibility is that NtPAE1 is also expressed in mature pollen grains and/or in growing pollen tubes, which was not analyzed in this work. Here, we suggest that the expression of the *NtPAE1* gene in stages preceding anthesis (Figure 1) may be related to an important role in the dissociation of cells in the secretory zone of the stigma and the transmitting tissue of the style, facilitating the formation of intercellular spaces, which are filled with exudate where pollen tubes grow [2]. A lower level of NtPAE1 activity in the stylar transmitting tissue would result in cells more closely adhered to each other, and this is exactly what we found in NtPAE1 silencing transgenic plants, in which the intercellular spaces are almost absent (Figure 6). Our results show the negative effects of NtPAE1 silencing on the growth of the pollen tube itself and on the maturation of the specialized tissues of the stigma/style, which do not reach the ideal state for supporting pollen tube growth (Figure 5). Conversely, plants with a higher level of NtPAE1 activity present a less structured transmitting tissue (Figure 6), also creating difficulties for pollen tube growth, probably due to growth with less proximity to the transmitting tissue cells. It is possible to speculate that this situation reduced cell-cell adhesion, which is important for efficient directional pollen tube growth. A similar situation was previously described, in which overexpression of *PtPAE1* in transgenic *N. tabacum* plants impaired the cellular elongation of floral styles and filaments, the germination of pollen grains, and the growth of pollen tubes [29]. Therefore, our results demonstrated that a well-regulated *NtPAE1* expression is crucial for the proper development of the transmitting tract.

Several studies have already demonstrated the importance of pectin modulation by PMEs in microgametogenesis and pollen grain germination [70,71,72,73]. In plant reproduction, PMEs have roles, for example, in pollen grain germination [18] and in regulating pollen tube growth [19,20,21]. Many esterases present in pollen grains are released upon contact with the stigma surface and during pollen germination [74]. Rejón and collaborators (2012) identified acetylesterase activity during pollen germination in olive (*Olea europaea*), a wet stigma species. The authors suggested that these enzymes are involved in pollen germination, pollen tube growth, and penetration of the stigma. So, despite the fact that we do not directly assay the expression of *NtPAE1* in pollen grains, its expression is expected in pollen grains and germinating pollen tubes. Accordingly, we demonstrated that pollen development is impaired in *NtPAE1*-silencing transgenic plants, resulting in malformed pollen grains (Figure 4), defective pollen germination, and pollen tube growth (Figure 5).

To our knowledge, this is the first work to assess the effects of reducing endogenous PAE activity, in contrast to evaluating ectopic expression [29,34]. Taken together, our results clearly established the importance of the well-regulated expression of *NtPAE1* for pollen and pistil development and, as a whole, the significance of pectin acetylesterase for successful plant reproduction in *N. tabacum*.

## Figures and Tables

**Figure 1 plants-12-00329-f001:**
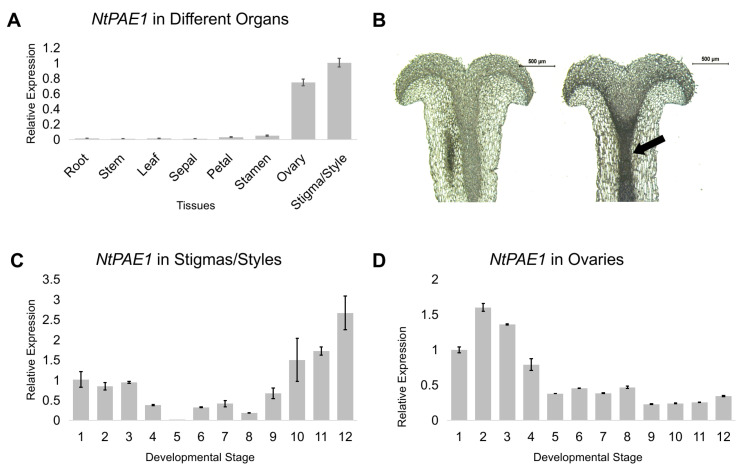
Expression of NtPAE1 in floral organs of tobacco. (**A**) Relative expression levels of *NtPAE1* by RT-qPCR in different tobacco tissues. (**B**) In situ hybridization of *NtPAE1* in stigma/style section. Left = sense *NtPAE1* probe; Right = antisense *NtPAE1* probe. Bars = 500 µm. (**C**) Relative expression levels of *NtPAE1* in stigmas and styles in different flower developmental stages (according to Koltunow et al. [38]). (**D**) Relative expression levels of *NtPAE1* in ovaries in different flower developmental stages.

**Figure 2 plants-12-00329-f002:**
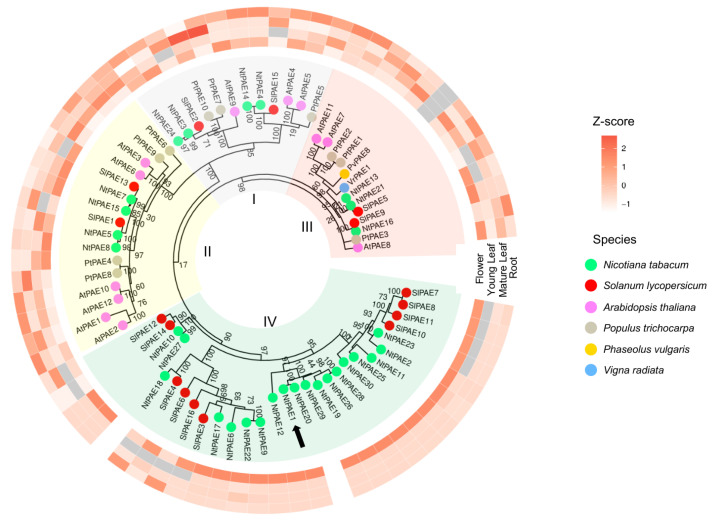
Maximum likelihood phylogenetic tree of PAE proteins and their expression levels in different plant tissues. The circles on the tree tips represent the species of each sequence, according to the legend on the right. Numbers close to the nodes represent the percentage of bootstrap replicates (out of 1000) that support them. Green-shaded highlight: clade IV, represented exclusively by Solanaceae PAEs, which are preferentially expressed in flowers. Black arrow: NtPAE1. Yellow-shaded highlight: clade II; Blue shaded highlight: clade I; Red shaded highlight: clade III. The expression levels of PAEs, represented as z-scores, are shown in the matrix above their corresponding PAE, for four different tissues—roots, young leaves, mature leaves, and flowers. Gray squares in the expression matrix represent missing data.

**Figure 3 plants-12-00329-f003:**
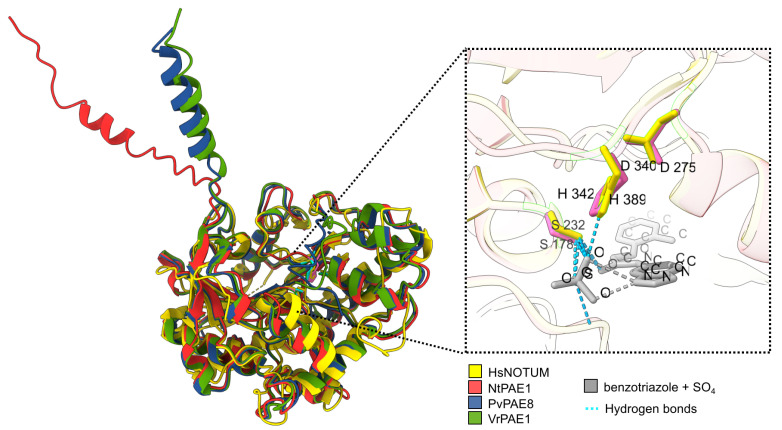
Plant PAEs are structurally similar to human NOTUM. Structural superposition of NtPAE1, PvPAE8, and VrPAE1 with HsNOTUM as a reference. The active site residues are structurally similar between all four proteins. The detail shows NOTUM active site residues (yellow) and putative active site of NtPAE1 (pink). The SO_4_ and benzotriazole molecules are ligands of NOTUM (PDB entry 6RQ8). NtPAE1 predicted active site residues = S178, D275, H342. HsNOTUM active site residues = S232, D340, H389.

**Figure 4 plants-12-00329-f004:**
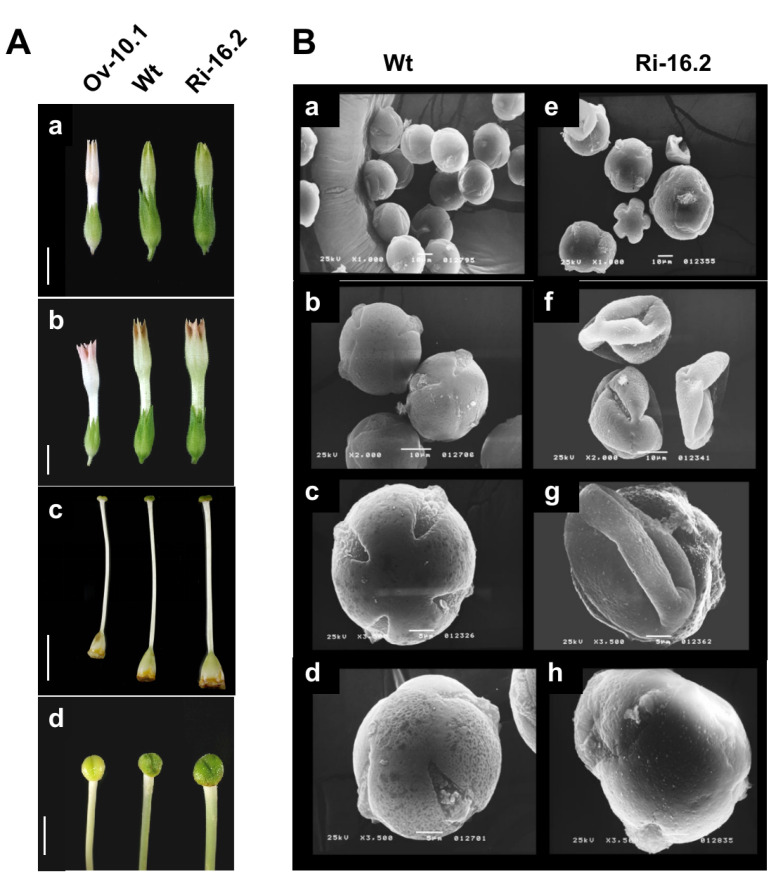
Morphological differences between wild type and transgenic plants. (**A**) Comparison of the phenotypes of transgenic and wild type plants. In all images, wild type plants (WT) are localized between transgenic plants PAEOv-10.1 and PAERi-16.2. (**a**) floral bud at stage 7. Bar = 1 cm; (**b**) floral bud at stage 10. Bar = 1 cm; (**c**) pistils of transgenic plants and wild type at stage 12; (**d**) Stigmas of wild type and transgenic plants. Bar = 3 mm. (**B**) Scanning electron microscopy of pollen grains of wild type (**a**–**d**) and PAERi-16.2 (**e**–**h**) plants. The pollen grains of PAERi-16.2 show irregular walls and apertures compared to the wild type.

**Figure 5 plants-12-00329-f005:**
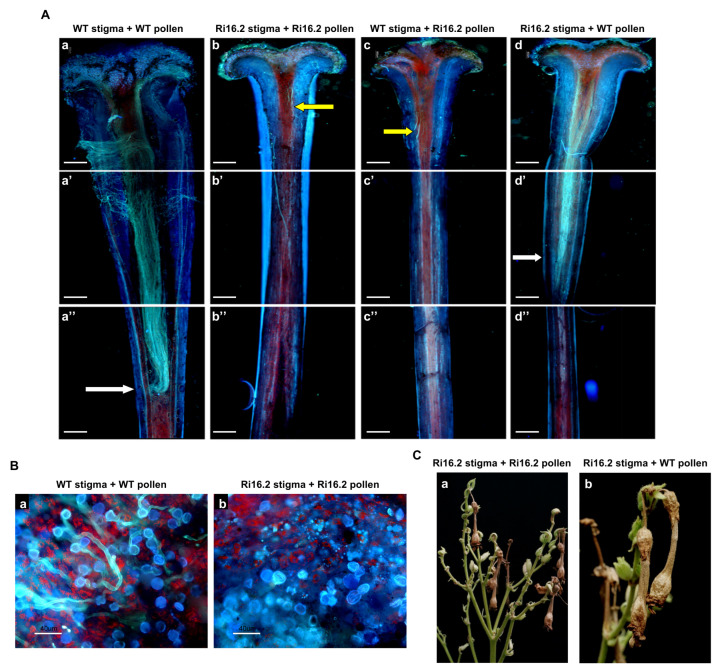
Microscopic analyzes show the impairment of pollen grains from plants with the silencing of NtPAE1. (**A**) Fluorescence microscopy of wild type and PAERi-16.2 (silencing) pistils at stage 11 of flower development stained with aniline blue. (**a**–**a”**) self-pollination of wild type plants; (**b**–**b”**) self-pollination of PAERi-16.2 silencing plant; (**c**–**c”**) cross-pollination of wild type stigma with pollen from PAERi-16.2; (**d**–**d”**) cross-pollination of PAERi-16.2 stigmas pollinated with wild type pollen. White and yellow arrows show the tips of the pollen tubes after 7 h of pollination. White bars = 800 µm. (**B**) Details of pollen tube growth in self-pollination of (**a**) wild type and (**b**) PAERi-16.2 plants. Bar = 40 µm. (**C**) Inflorescence and fruits of PAERi-16.2 transgenic plant. (**a**) Self-pollinated; (**b**) cross-pollinated with wild type pollen.

**Figure 6 plants-12-00329-f006:**
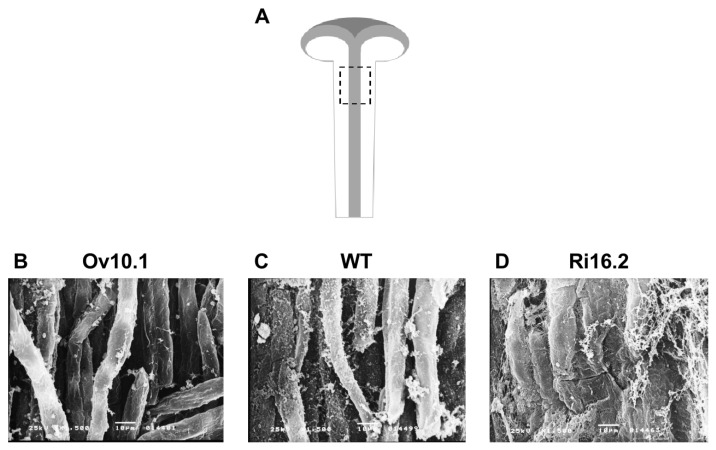
Scanning electron microscopy (SEM) of stage 11 stylar transmitting tissue of wild type (SR1) and transgenic plants.(**A**) Drawing of stigma/style, the dotted line box indicates the portion analyzed by SEM. (**B**) SEM of the STT from line Ov10.1 transgenic plant overexpressing *NtPAE1*. (**C**) SEM of the STT from a wild type plant. (**D**) SEM of the STT from line Ri16.2 transgenic plant silencing *NtPAE1*. The images highlight the differences in the cell-cell adhesion on the first one-third of the stylar transmitting tissue of transgenic plants in relation to the wild type plant (1500× magnification). Bar = 10 µm.

## Data Availability

The data presented in this study are available within the article and as Appendix A. The nucleotide sequence of *NtPAE1* is deposited in GenBank under accession OQ091254. Clones and materials studied are available upon request from the corresponding authors.

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
