# Peer review of "Silencing of a Pectin Acetylesterase (PAE) Gene Highly Expressed in Tobacco Pistils Negatively Affects Pollen Tube Growth"

_plants, 2023, doi:10.3390/plants12020329_

Round 1
Reviewer 1 Report
The presented manuscript on a Pectin Acetylesterase (PAE) highly expressed in tobacco pistils and with potential important functions in plant reproduction is very well presented and the experiments well designed.
Since there are not so many studies on PAEs and its involvement in reproduction, this makes this MS of high interest to the experts working in this field of study.
It would be interesting, in further studies on this PAE, to perform immunolocalization studies for different types of pectins in the RNAi lines, to search for possible differences with the WT plants.
Also, regarding the abnormal pollen grains, which seems to me aborted pollen, it would also be a nice add on to perform a test to check for their viability, such as the Alexander staining. I hope the authors will continue to study this gene in the future, it might bring some more interesting results.
Below I attach a PDF with some minor comments for the author to address.

Reviewer 2 Report
Evaluation of the genes involved in plant ontogeny is one of the stable trends in modern biology. This study is of a fundamental nature, but since it affects the reproductive sphere, a practical outcome can also be traced.
Deciphering the development of programs, as well as the study of the molecular genetic mechanisms underlying the regulation of ontogenesis, is a progressive task, which manifests itself in the study of the genetic foundations and physiological regulation of the development of plant reproduction. For this reason, this experiment is a relevant and intended contribution to understanding the developmental characteristics of individual plants.
The applied methods are adequate and modern. The results are clear and objective. However, there are some minor remarks that need to be noted.
1. Line 112... The authors write that: «Samples for RNA extractions were collected from individual plant organs and 12 floral developmental stages…”. A clarification is required. What are the organs?
2. The authors study 12 stages of flower development. References to work where they are described [38] are not enough. It is necessary to describe all 12 stages in the methods.
3. The authors use in the article abbreviated Latin specific names. Full names should be given, at least when they appear for the first time in the text.
4. The photographs of pollen and pistil obtained on a scanning electron microscope are very adorn the work. Why is figure S6 not included in the text of the article?
5. In the method of staining with aniline blue, it is not clear how the pistils were prepared for staining. Did you cut the pistils? Or was maceration carried out and crushed preparations were prepared? This may be the reason why the photos b, c in figure A5 are not convincing.
Based on the mentioned above, I think that this article can by recommended for publication in the «Plants» after revision.
